# Aqueous habitats and carbon inputs shape the microscale geography and interaction ranges of soil bacteria

Samuel Bickel [1,2✉] & Dani Or [1,3]

Earth's diverse soil microbiomes host bacteria within dynamic and fragmented aqueous habitats that occupy complex pore spaces and restrict the spatial range of ecological interactions. Yet, the spatial distributions of bacterial cells in soil communities remain underexplored. Here, we propose a modelling framework representing submillimeter-scale distributions of soil bacteria based on physical constraints supported by individual-based model results and direct observations. The spatial distribution of bacterial cell clusters modulates various metabolic interactions and soil microbiome functioning. Dry soils with long diffusion times limit localized interactions of the sparse communities. Frequently wet soils enable long-range trophic interactions between dense cell clusters through connected aqueous pathways. Biomes with high carbon inputs promote large and dense cell clusters where anoxic microsites form even in aerated soils. Micro-geographic considerations of difficult-to-observe microbial processes can improve the interpretation of data from bulk soil samples.

[1] ETH Zurich, Zürich 8092, Switzerland. [2] Graz University of Technology, Graz 8010, Austria. [3] Desert Research Institute, Reno, NV, USA. ✉email: samuel.
bickel@tugraz.at

The focus on large-scale biogeographic patterns[1–6] has revealed important drivers for soil microbial abundance[2,3] and diversity[1,4]. Soil bacteria rank high in the global biomass distribution[6] and provide crucial ecosystem functions[4,7,8]. Interactions between soil properties, precipitation[3], temperature, and associated factors such as vegetation-derived primary productivity[2,5] exert significant control over macroscopic measures such as bacterial abundance or diversity. Yet, how bacterial populations are spatially organized in soil pore spaces and how they interact at small scales relevant to their life strategies[9–13] remain largely unknown. Elucidating typical ranges of bacterial interactions at the cell[14] or colony scales as shaped by their physical environment can benefit the interpretation of measurements made at coarser scales (bulk samples or soil profiles) and provide mechanistic insights into the functioning of the soil microbiome in different biomes[15–18].

Evidence suggests that bacterial cells are spatially aggregated[9] and exhibit highly localized activity in soil[19,20] and marine sediments[21]. Complex pore spaces with a large specific surface area are characteristic of soil bacterial habitats[22], yet bacterial cells occupy less than 1% of the available surface area[13,23]. Soils of temperate regions host bacterial cell densities between $10^7$ and $10^{10}$ cells per gram of soil[9]. Bacterial cells tend to aggregate near living plant roots[24] or the active rhizosphere[25], and in the detritusphere[26] around plant-derived particulate organic matter (POM). Bioturbation and root growth are the primary mechanisms for introducing POM into the soil. The soil aqueous phase connectivity shapes diffusive transport, cell dispersion, and access to patchy resources[22,26]. In water-replete environments, soil bacteria may attain cell densities similar to those found in biofilms[12,27] that host diverse communities with structural properties reminiscent of urbanization[27]. A few dense bacterial "megacities" may constitute a substantial proportion of the bacterial biomass. The rest of the soil bacterial population is distributed across numerous "settlements" containing 10–100 cells each[9]. Many standard soil microbiome analyses use centimeter-scale bulk samples that potentially mix spatially isolated and distinct populations[15,19]. This limits the attribution of soil microbiome functioning to properties such as bacterial biomass, diversity, and bulk activity.

Although direct observations in soil are limited, we expect that the vast number of soil bacterial species[4] and their complex biochemical functions[7,16] are spatially distributed and interact via dynamically connected aqueous microhabitats[17]. Processes such as the development of nanotube infrastructure[28], electron transport[29], gene transfer[14], and cell–cell signaling[30] require proximity among bacterial cells. The aqueous phase surrounding these cells facilitates diffusion of substances (e.g., metabolites, antibiotics, quorum sensing) and likely constrains the expression of various general traits[20,26,31]. The spatial distribution of soil POM and average aqueous-phase connectivity jointly affect ecological processes that involve resource and metabolite exchanges between bacterial populations[16,17]. Characteristic diffusion times for mediating metabolic interactions can vary from seconds to months between wet and dry soils, respectively (Supplementary Note 1). Thus, distances across which soil bacterial cells can effectively exchange metabolites and information modulates the functioning of spatially distributed bacterial communities in soil[18]. Despite this critical aspect of bacterial life in soil, we lack understanding regarding of bacterial interactions and exchanges embedded within complex soil microarchitecture.

Here, we introduce a quantitative framework that considers distinct levels of resource availability and soil aqueous phase connectivity to estimate effective spatial ranges of soil bacterial interactions. Limitations to direct observations of soil-bacterial distributions and interactions at the microscale, serve as the impetus to advance mechanistic biophysical models that bridge this knowledge gap and can provide insights into soil bacterial micro-geography. We employ a spatially explicit individual-based model (SIM) that simulates bacterial growth on hydrated soil surfaces to obtain spatial distributions of bacterial cells. The specific objectives of this study were: (i) to quantify the spatial variation of soil bacterial cell density based on biome-specific soil carrying capacity and aqueous phase connectivity; (ii) to link bacterial cell cluster size distribution to sample-scale (bulk) cell density; and (iii) to demonstrate how cell cluster size variations could affect bacterial interactions and soil ecosystem functioning.

Supported by few observations and individual-based model results, we propose a *bacterial interaction heuristic model* (BIHM) that relies on well-established aggregation statistics of ecological populations[32–34] to predict the spatial distributions of soil bacterial communities as a function of auxiliary variables. The BIHM is an analytical formulation (Eqs. 12, 14, 19, and 20) that links biome-specific bacterial cell density determined by carbon inputs with cell cluster size distributions across different climate conditions and soil types. Two important processes in unsaturated soil are embedded in the BIHM: the dependence of bulk cell density on soil carrying capacity where access to patchy resources is mediated by diffusive transport[5], and the natural spatial aggregation of bacterial cells due to local cell division and growth under constrained dispersal ranges[22]. At the centimeter scale (macroscale), the templates that govern soil bacterial communities and their interactions[31] vary with resource abundance that shapes bacterial bulk cell density across different biomes (Fig. 1a). We link bulk cell density with microscopic bacterial cell cluster size distributions at sub-millimeter scales using aggregation statistical laws to connect biome-specific bacterial micro-geography with soil microbiome functioning.

An important implication of bacterial cell aggregation in soil is the emergence of large cell clusters at locations with limited cell dispersal and high growth. Near resource patches, clusters of sessile cells can grow into sizable colonies in which resource gradients establish due to cellular activity. For example, anoxic microsites[35] can arise spontaneously where diffusion-limited oxygen fluxes are depleted in the core of dense and compact bacterial cell clusters[36]. The amount of biomass associated with putative anoxic microsites may vary with soil type[35] and bacterial population sizes[35,36] across environmental conditions and biomes.

## Results

### Soil bacterial cell density and cell cluster size distribution linked to rainfall and vegetation. 
Nearly 35% of a biome's net primary productivity (NPP) is partitioned into new fine roots[37] that contribute to soil POM's annual turnover[38]. Around one-quarter of this belowground NPP feeds soil bacterial biomass[39,40] at densities that decline with soil depth[11] following the vertical distribution of plant roots and other conditions in soil profiles[2]. On average the distance to a source of POM decreases with increasing NPP (Fig. 1b) while the length a small molecule could diffuse for one year increases with rainfall frequency (Fig. 1c). Hence, the number of bacterial cells maintained around sources of POM can be calculated for a range of rainfall frequencies using information on fine root fragments and estimated diffusive distances. The average distance across which bacterial cells can intercept diffusing nutrients depends on the aqueous phase connectivity of unsaturated soils[41] shaped by the time interval between rainfall events. Using a biomes' mean annual NPP, we assumed that local bacterial biomass decays exponentially around each POM nucleolus with a characteristic distance defined by the effective diffusive length for a given soil type and climatic

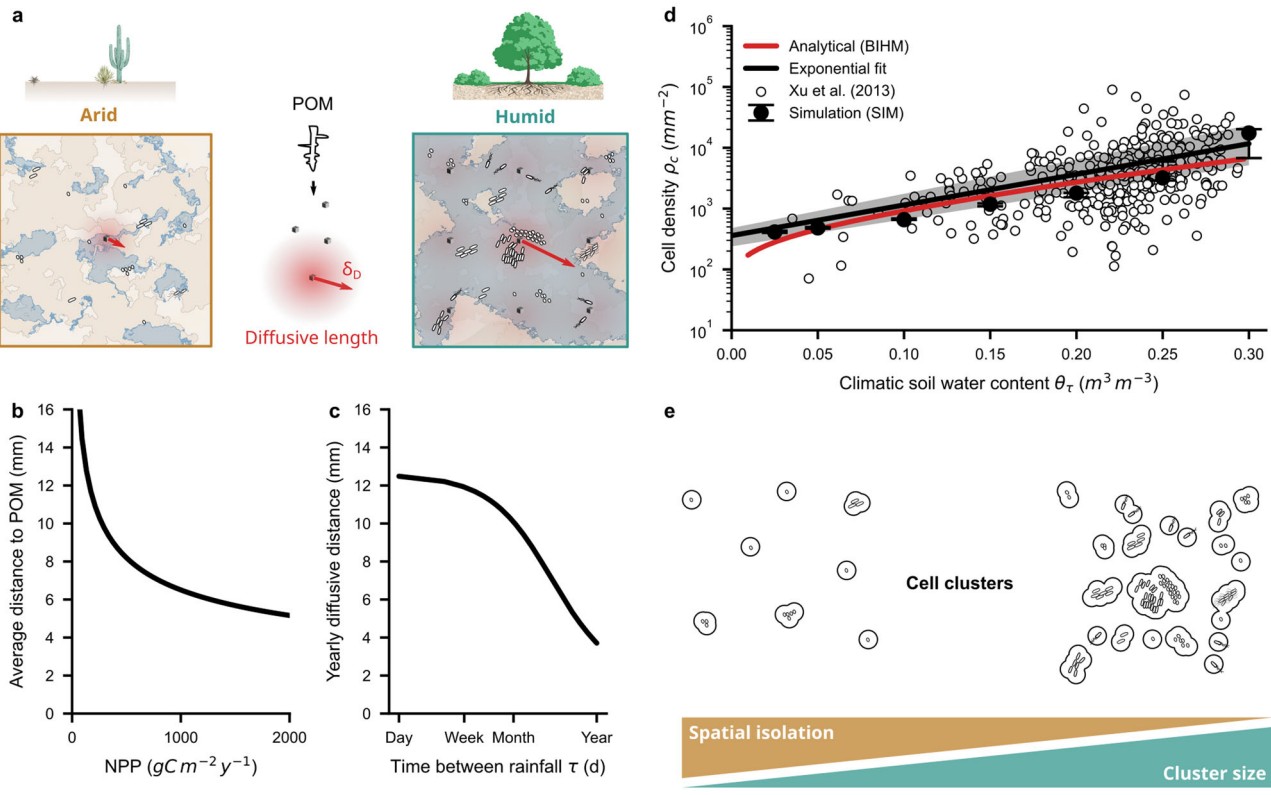

**Fig. 1 Soil bacterial cell density and cell cluster sizes across biomes. a** Particulate organic matter (POM) derived from net primary productivity (NPP) provides the carbon input to soil bacteria. **b** The average distance to POM depends on NPP. In arid soil, the sparse distribution of POM supports only a few bacterial cells. **c** Diffusive fluxes increase in humid environments with higher soil water connectivity and higher NPP. The distance that resources could travel via diffusion depends on the time between rainfall events. **d** Bacterial cell density on soil surfaces increases with climatic water content. Symbols indicate data of topsoil microbial biomass carbon[2] converted to estimates of bacterial cell density (black line - exponential fit, $n = 429$). The bacterial interaction heuristic model (BIHM, red line) considers soil properties, climate, and vegetation to estimate the average cell density that is compared with the SIM (median ± IQR, $n = 9$). **e** Localized growth, and motility aggregate soil bacterial cells at the microscale. Enhanced interaction ranges, increased cluster sizes, and reduced spatial isolation characterize well-connected environments.

condition (Eqs. 8–10). The resulting bacterial bulk cell densities are described as a function of average soil and climatic conditions using the BIHM (Fig. 1d). For comparison, previously reported[2] microbial biomass carbon (topsoil, $n = 429$) has been converted[5] to estimates of bacterial bulk cell density (Fig. 1d). Independent estimates of bacterial bulk cell densities were obtained from the SIM that simulates growth and dispersal of individual bacterial cells living on hydrated soil surfaces at the submillimeter scale[5]. Both, numerical results (SIM) and global bacterial abundance data[2] indicate a disproportionate increase of bacterial bulk cell density with increasing soil water content (Fig. 1d). This average (macroscopic) cell density shapes the cell cluster size distribution on soil surfaces (Fig. 1e).

**Model predictions of bacterial cell cluster size distributions in soil.** We present mechanistic simulations of bacterial populations using the SIM supplemented by examples of observed spatial cell distributions from a microcosm experiment conducted for this study. Bacterial cells were assigned to a cell cluster if they were located within five micrometers distance to neighboring cells[34]. In the microcosm experiment we observed cell cluster size distributions across different hydration and nutrient conditions (Fig. 2a). Although, we did not observe effects of the treatment on the distribution of cell cluster sizes, the simulations by the SIM had substantial variation in bacterial cell cluster sizes with changes in hydration conditions and bacterial cell densities (Fig. 2b). The available information from simulation results, soil

observations[9] and our microcosm experiments indicate that soil bacterial cell cluster sizes could be described by an exponentially truncated power law. This observed pattern was not assumed in the BIHM a priori and was evidenced by simulations and experiments (Fig. 2). Similar spatial aggregation patterns have been previously observed for bacteria and other organisms[5,32–34]. Thus, we accepted this as a tentative representation and estimated the exponent $b$ and cutoff size $n_c$ (Eq. 14) to quantify the size distribution of bacterial cell clusters. The observed cell cluster size curves collapse onto a single relationship by rescaling with the obtained parameters ($b$ and $n_c$) thus lending support for the proposed cell cluster size distribution model (Eq. 14) that describes experimental data and simulation results (Fig. 2c, d). We note that the SIM makes no assumptions regarding the bacterial cell cluster size distribution.

Two examples of the observed spatial cell distributions are shown for the experiment and simulations (Fig. 3a, b, respectively). The cell cluster size distributions vary with the total number of individual cells, as deduced from direct measurement of cell density[33,34]. This might explain why we could not observe clear differences between the experimental treatments, which resulted in similar average cell densities and cluster sizes (Fig. S1). Hence, we used the microscopic cell cluster size distributions obtained from the SIM to parametrize the BIHM (Eqs. 19 and 20; Supplementary Note 2) that estimates soil cell densities based on rainfall frequency, soil type and carbon input (Eqs. 1–12), and considers the spatial aggregation of bacterial cells as a function of bulk cell density (Eqs. 14–20). The

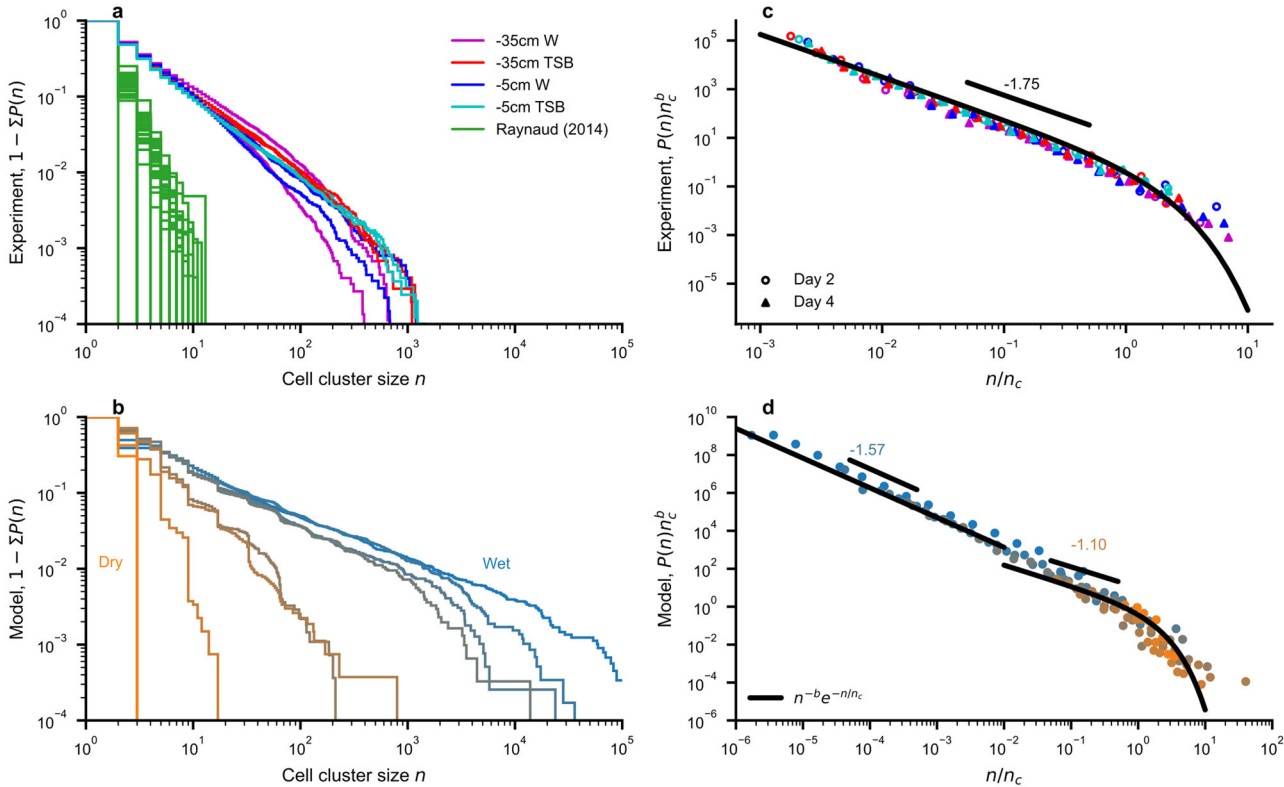

**Fig. 2 Observed and predicted cell cluster size distributions. a** and **b** show the complementary cumulative probability $1 - \sum P(n)$ of observing a cluster with $n$ cells aggregated within 5 μm. **a** Cluster size distributions from two days of microcosm measurements (purple, red, blue, and cyan) under varying hydration (−35 cm and −5 cm matric potential) and nutrient conditions (tap water W and tryptic soy broth TSB). Soil thin-section data from an independent study[9] is shown in green. **b** The distributions obtained for a range of water contents from a spatially explicit individual-based model (SIM). **c** and **d** show rescaled $P(n)$ using obtained parameters ($b$ and $n_c$). **c** The microcosm data collapse after rescaling with an average exponent ($b \approx 1.75$). Round symbols and triangles indicate measurements of day two and day four, respectively. **d** Distributions obtained from the SIM with different exponents above and below water content of 0.2 ($b = 1.57$ and 1.10 for wet and dry, respectively).

resulting bacterial cell cluster size distributions are illustrated for high and low bulk cell densities (Fig. 3c). For comparison, we included data from a previous study[9] using soil thin sections with low average cell density (around 500 cells per mm²; $n_{images} = 341$, $n_{cells} = 46,151$) and our microcosm experiment using nutrient-rich garden soil with high average cell density (around 20,000 cells per mm²; $n_{images} = 90$, $n_{cells} = 640,100$). These observed cell cluster size distributions lend support to the BIHM results and demonstrate the transition from an exponential to a power law distribution at high cell densities. We implemented the BIHM to estimate the proportion of bacterial biomass associated with small and larger cell clusters (>100 cells) across a range of climatic water contents assuming constant mean annual temperature (MAT) and mean NPP (Fig. 3d). Like the well-known aridity index, climatic water content is a proxy variable that considers soil water holding capacity, rainfall frequency and potential evaporation[5]. The enhanced soil carrying capacity of humid environments with high carbon inputs and low temperatures[5] supports the proliferation of large bacterial cell clusters.

**The distribution of bacterial cell cluster sizes shapes the strength of metabolic interactions in soil**. To quantify the strength (or ranges) of diffusion-mediated metabolic interactions among different soil bacterial communities, we estimated average distances between bacterial cell clusters emerging for different cluster sizes and climatic water contents (Fig. S2). Congruent with the assumption of spatially uniform POM distribution within a thin slab of soil at a given soil depth, the distance between cell

clusters in a soil is also assumed uniform. This simplification facilitates the use of volume-averaged macroscopic quantities such as effective nutrient diffusivities and carbon input fluxes while preserving microscopic variations in cell cluster sizes and numbers. The average distance between clusters containing at least two cells was about 100 μm and did not vary much with soil wetness considering average NPP, MAT, and soil type. However, the distance between larger clusters (>100 cells) increased rapidly with reduced soil water contents. To quantify the extent of temporal separation between cell clusters, we estimated how long it would take a small molecule to diffuse across the average inter-cluster separation distance (Fig. S2). This timescale increased from hours to months as the soil became drier affecting the distribution of shared resources and limiting the ranges for cell–cell interactions[14].

We have used the mechanistic individual-based SIM to study how soil bacterial interactions depend on the distribution and connectivity of aqueous habitats and carbon inputs. For simplicity, the SIM considers a conversion of three substrates ($A \rightarrow B \rightarrow C$) by cells of different species ranging from specialists to generalists (that may use between one to three substrates, respectively). Metabolic interactions between distinct species are based on exchanging substrates via diffusion. Thus, interactions are suppressed under dry conditions with implications for bacterial community composition and diversity[5]. The SIM considers that substrate $A$ is initially supplied from a point source and diffusive fluxes vary in space depending on aqueous phase connectivity. Simulation results show how conversion from the supplied substrate $A$ to the end-product $C$ depends on water

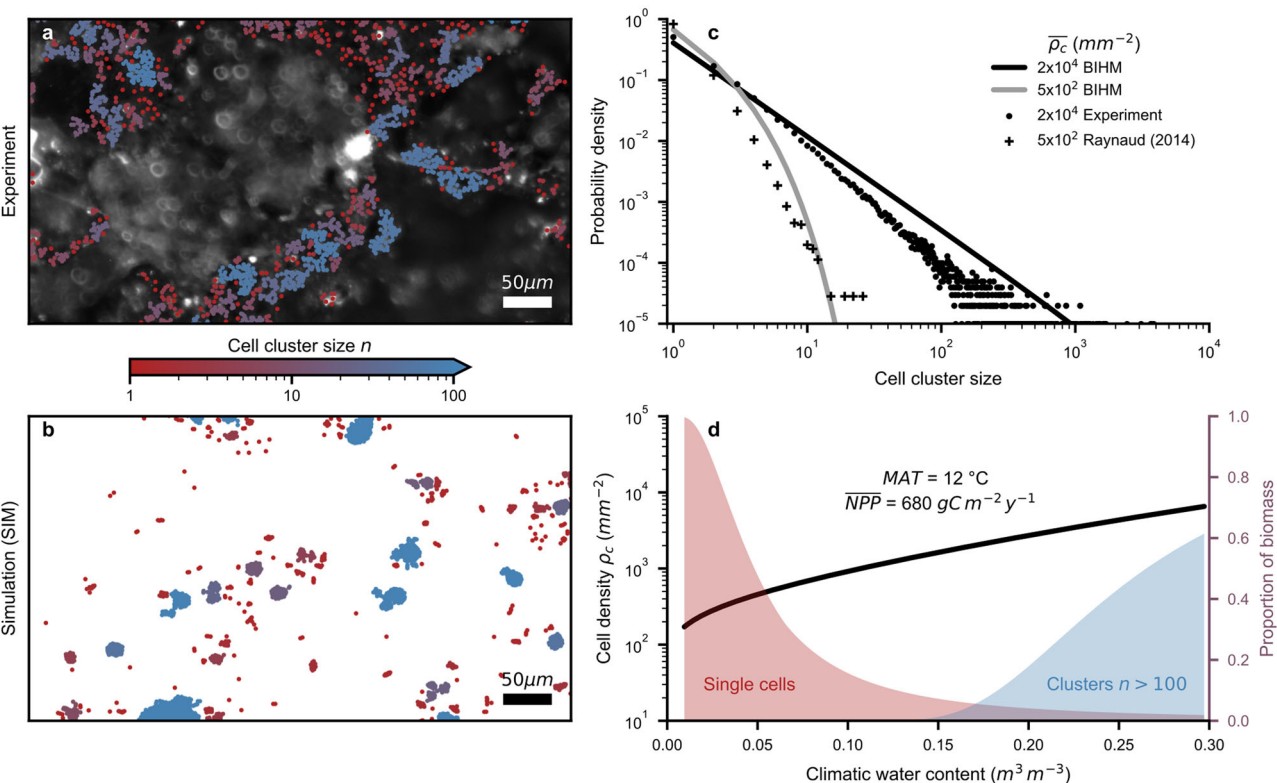

**Fig. 3 Soil bacterial spatial distribution and cell cluster sizes vary with cell density. a** Example of experimentally detected bacterial cell clusters on soil surfaces. Only the regions in focus were analyzed (SYTO9 intensity in greyscale). Colors indicate cluster sizes (cell number, $n$). **b** The spatial distribution of cells obtained from a spatially explicit individual-based model (SIM). **c** The predicted soil bacterial cell cluster size distribution is shown for two average cell densities using the parametrized BIHM. Data from our microcosm experiments (circles; $n_{images} = 90$, $n_{cells} = 640,100$) and a previous study[9] (crosses; $n_{images} = 341$, $n_{cells} = 46,151$) with high and low cell densities support the BIHM. **d** Modeled cell density as a function of climatic water content for mean annual temperature (MAT) and mean annual net primary productivity (NPP). The second axis shows the predicted proportion of bacterial biomass associated with single cells (red) and clusters with more than a hundred cells (blue).

contents (Fig. S3a). Although enhanced interactions and metabolite exchanges under wet conditions enabled higher bacterial cell densities and species richness (Fig. S3b), the Shannon index decreased towards higher water contents. This highlights how community dynamics are affected by spatial partitioning under predominantly competitive interactions[18]. The predicted bacterial richness increased with the total number of cell clusters while community evenness decreased towards wet conditions associated with higher resource fluxes[5,31] and stronger interspecific interactions (Fig. S3c).

**Large bacterial cell clusters may induce anoxic microsites across a wide range of soil hydration conditions.** The distribution of cell cluster sizes follows a truncated power law that naturally emerges from the SIM, which makes no assumptions about the positioning of cells. The cell cluster size distribution is more skewed in wet soils, which have a higher carrying capacity and a higher cutoff size. This implies that the largest bacterial cell clusters (megacities) become bigger and smaller cell clusters (settlements) become more numerous towards humid biomes with high carbon inputs. Thus, the few bacterial megacities may disproportionally affect cell-cell interactions and ecosystem functioning. A potentially important implication of the spatial distribution of bacterial cell cluster sizes is the emergence of anoxic microsites within hotpots of bacterial activity. Activity within compact (and dense) cell clusters can result in the depletion of oxygen fluxes supplied by diffusion via soil pores whenever cell respiration rates exceed diffusive supply rates within a cell cluster[36] (Fig. 4a). For simplicity, we considered fully

packed ($10^9$ cells per mm$^3$) spherical cell clusters embedded in unsaturated soil to estimate the minimum cluster radius required to induce an anoxic core[36]. Assuming oxygen concentrations in the soil liquid phase reflect equilibrium with atmospheric levels (a conservative assumption for many humid regions with considerable root respiration), we quantified the proportion of biomass associated with large cell clusters that could induce anoxic conditions as a function of cell density (Fig. 4b). Using spatially distributed soil properties[42,43], climate attributes[44,45], and vegetation carbon input[46] at 0.1° resolution[5], we applied the BIHM to estimate the proportion of biomass associated with anoxic cell clusters (Fig. 4c). We removed permafrost soils[47] from the analysis due to stringent limitations on substrate diffusion not currently considered. The dominant land cover type[48] for 2019 was used to compare the amount of biomass associated with anoxic cell clusters for different biomes and land use (Fig. 4d). The number of anoxic cell clusters per gram of soil increased from 15 in bare soil to over 5500 in closed forests. The predicted amount of bacterial biomass in anoxic cell clusters tracked the bulk cell density and was highest for closed forests followed by herbaceous wetlands. For reference, we show previously reported cell counts of anaerobic species from soils of different biomes[49]. The values ranged between $10^4$ and $10^7$ cells per gram of soil and were lower than our model estimates. This was expected since not all cells of an anoxic cluster must belong to an anaerobic species. Another study reported cell counts of anaerobic species between $10^8$ and $10^{10}$ per gram of soil associated with plant residues in a rice paddy field[50] and marks an upper bound for very wet and organic matter rich soils. The importance of our estimate is in the

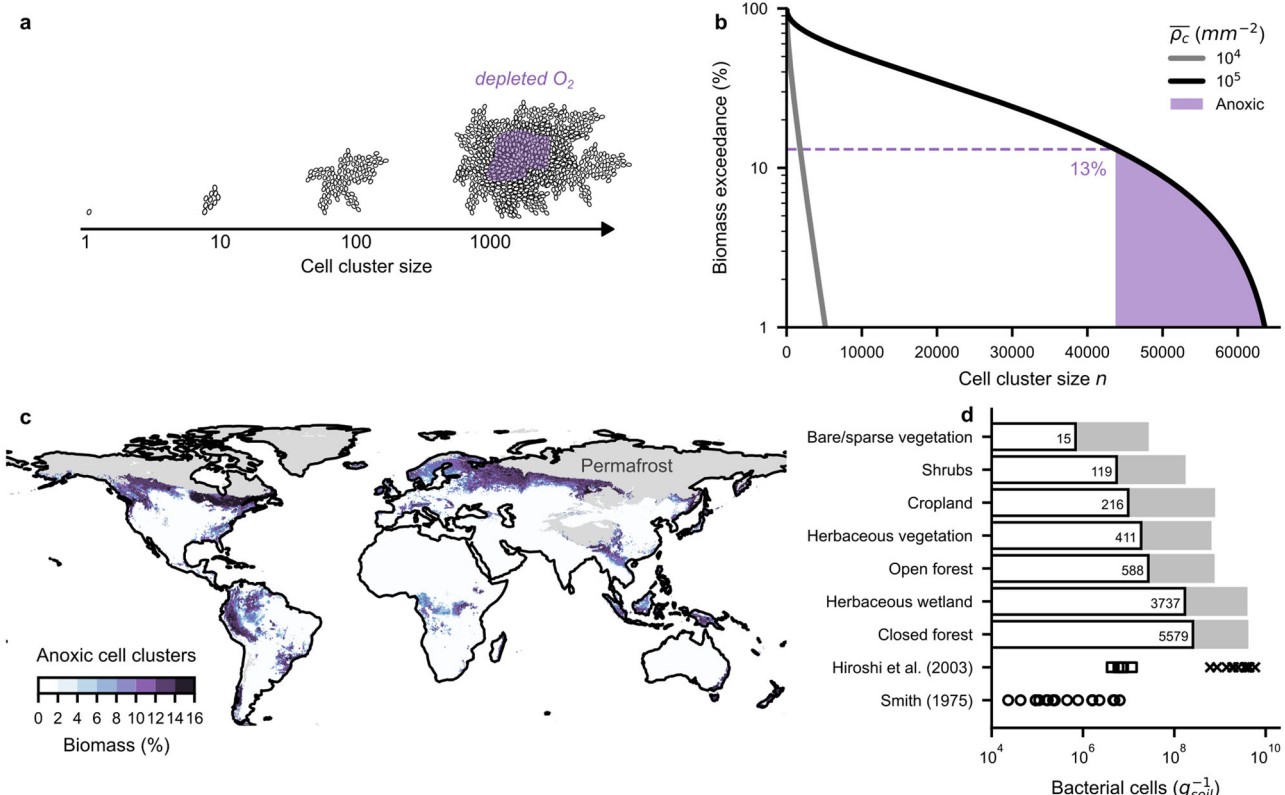

**Fig. 4 Regions with likely occurrence of anoxic microsites due to large soil bacterial cell clusters. a** Large cell clusters may spontaneously deplete oxygen in their core. **b** The percentage of biomass associated with cell clusters exceeding size *n*. The proportion of large cell clusters that promote anoxic conditions increases with cell density. **c** The predicted percentage of bacterial biomass associated with anoxic cell clusters globally. Permafrost soils were excluded from the analysis. **d** The area-weighted average number of bacterial cells in anoxic cell clusters for different land cover types. Gray bars indicate total cell density estimates. Numbers on bars indicate the number of anoxic clusters. Data from two studies are shown for comparison. The number of anaerobic bacteria in various soils is reported by Smith[49] (circles, $n = 21$). Hiroshi et al.[50] report cell densities of anaerobic bacteria for soil (squares, $n = 5$) and plant residues (crosses, $n = 12$) of a rice paddy field.

ubiquity of anoxic microsites even in aerated unsaturated soils, and the link to bacterial cluster size distribution that depends on macroscopic properties of the soil and its microbiome.

## Discussion

The highly dynamic soil aqueous phase connectivity limits dispersal ranges[22] that, in turn, affect the spatial arrangements of bacterial cells and their traits[16,17]. Irrespective of the specific mechanisms, bacterial biomass is not uniformly distributed in soil[9,10] with important implications for metabolite exchanges[7] and the onset of anaerobic respiration[17,36]. The spatial distribution of soil bacterial cells emphasizes the localized nature of interactions relevant to soil ecological functioning[11,16]. The size of cell clusters marks the extent of contact-dependent interactions within bacterial communities including direct cell-cell signaling[30], gene transfer[14], electron transfer[29], and morphological development[27,28]. It is useful to note, however, that most of the bulk soil volume is inhabited by small bacterial cell clusters[9] with unknown spatial separation and limited interactions with neighbors.

The connectivity of water on soil pore surfaces and the size of associated bacterial habitats shape resource fluxes that affect community diversity[5,31] and functioning[18]. Even in water-replete marine sediments, a large fraction of bacteria is attached to surfaces[12,51] with a few hot spots that display distinct species abundance distributions compared to those sampled from background communities[21]. Across terrestrial biomes, the variations in bulk cell density[2,3,5] define the spatial extent of bacterial

neighborhoods that interact via exchanges of metabolites. The reliance on diffusion-mediated processes for signal and metabolite exchanges offers quantifiable links between the physical micro-geography and the functional capacity of soil bacterial communities that interact via water "highways" connecting richer and larger bacterial cities.

Although more direct observations are needed to validate the proposed cell cluster size distributions and their relations with bulk cell density across soils of different biomes, the quantification of soil bacterial micro-geography is a critical step toward deciphering the complexity[17] of bacterial habitats. By incorporating the microscale spatial context of soil microbiome functioning[4,7,16,18] our tentative spatial distributions quantify ecological interactions beyond descriptions of well-mixed bacterial communities. The assumption of spatially uniform abundance at the sample scale that often underlies inferences of species interactions and co-occurrence requires careful evaluation as it is likely to bias[15] the picture of bacterial life under common soil conditions. The tentative estimates presented here provide a tractable modeling approach that is based on only few basic biophysical processes.

It is noteworthy that the bacterial bulk cell densities presented in our heuristic model (BIHM) reflect average soil conditions and modifications for environments with different controls over connectivity and carbon supply could be introduced (e.g., for biocrusts[20] or the immediate rhizosphere[24]). Refinements to the model might consider stoichiometric limitations on carrying capacity and the use of distributed bacterial trait values (e.g.,

oxygen uptake rates). Interactions with other soil microorganisms such as the competition with fungi[4] could further affect bacterial cell density in ecosystems where fungi are most prevalent[39] (e.g., in forests). The BIHM provides a parsimonious and general basis for considering interactions of soil bacteria with other organisms relevant to soil ecosystem functioning. We could interpret bacterial cell clusters as spatially distributed hosts and foraging grounds for bacteriophages and soil fauna, respectively.

The perspective presented here offers a tentative, yet unifying, framework for linking soil bacterial cell cluster sizes to spatial[14,28–30] and metabolic[20,29,31] interactions based on sample-scale cell density. The few large cell clusters that might develop anoxic cores[36] are restricted to densely populated resource patches that can be decoupled from oxygen (and redox) conditions of the bulk soil[35]. A small increase in soil anoxic volume can greatly reduce carbon mineralization[52] and affects soil gaseous emission from the microscale[19,20]. The large disparity in cell cluster size distributions predicts that there will be only a few anoxic cell clusters under a wide range of soil conditions that drive the persistence of soil carbon[16,52] and associated greenhouse gas emissions across biomes[8].

## Materials and methods

### Average cell density based on diffusion and distance to particulate organic matter (POM).
In the bacterial interactions heuristic model (BIHM), the maximal number of cells maintained in a soil volume ("soil carrying capacity") is linked to carbon input by vegetation and cell-specific maintenance rate that is sensitive to temperature[5]. Bulk cell density is estimated based on yearly averaged net primary productivity ($NPP$) that enters a section of the soil profile as new roots[37] ($\xi = 0.35$) and is available to soil bacteria[39,40] ($NPP_{b,z}$, with $\epsilon = 0.24$). The vertical distribution of carbon $f(z)$ to a maximum soil depth ($d_{soil} = 1$ m) is described using a log-normal distribution with $\mu = 0.18$ and $\sigma = 1.00$ as previously reported[5] (Eq. 1). We consider a homogeneous topsoil to a depth of 0.1 m for integration.

$$NPP_{b,z} = \xi \epsilon \frac{NPP}{d_{soil}} F_z = \xi \epsilon \frac{NPP}{d_{soil}} \int_0^{0.1} f(z)dz \tag{1}$$

The bulk cell density at carrying capacity $\rho_{CC}$ is estimated (Eq. 2) by assuming bacterial cells with mass $M_c = 10^{-13}$ gC, maintenance rate $m = 1.5$ gC gC$^{-1}$ y$^{-1}$ and temperature sensitivity[5] $f_T$ as previously described[5].

$$\rho_{CC}(z, T) = \frac{NPP_{b,z}}{f_T m M_c} \tag{2}$$

We assume that the main source of carbon for soil bacteria is POM (and exudation) derived from fine roots with a turnover time of one year[38]. The yearly average volume of POM ($V_{POM}$) was estimated based on $NPP_{b,z}$ and the density of fine roots[38] $\rho_{FR} = 0.5$ g cm$^{-3}$. The yearly number of POM fragments $N_{POM}$ is estimated based on a fine root diameter[38] $d_{FR} = 0.5$ mm. Assuming POM has an uniform spatial distribution at the centimeter scale, we calculate an average distance to POM $\delta_{POM}$ (Eqs. 3–5).

$$V_{POM} \cong \frac{NPP_{b,z}}{\rho_{FR}} \tag{3}$$

$$N_{POM} = \frac{V_{POM}}{d_{FR}^3} \tag{4}$$

$$\delta_{POM} = \left(\frac{V_{soil}}{N_{POM}}\right)^{\frac{1}{3}} \tag{5}$$

The (climatic) soil water content $\theta$ is defined as previously reported[5] (Eqs. 6). The model assumes evaporation from the soil after drainage to field capacity $\theta_{FC}$ (approx. half of the water content at saturation $\theta_s$). The soil is left for drying over given time $t$ with a constant rate $\alpha$ (estimated using potential evapotranspiration $PET$). A climatic average timescale $\tau$ over which the soil dries can be estimated as the number of consecutive dry days[5] using a precipitation time series[45]. The time between rainfall events during which the soil is wet is used to calculate the average number of wetting cycles per year $N_{cyc}$ (Eq. 7).

$$\theta = \theta_{FC} e^{-\alpha t} \text{ with } \alpha = \frac{PET}{d_{soil} \theta_{FC}} \text{ and } \theta_{FC} \cong \frac{\theta_s}{2} \tag{6}$$

$$N_{cyc} = \frac{365}{\tau} \tag{7}$$

The total distance a small molecule could travel during a year when released from a point source of POM is related to soil effective diffusivity[41] (Eq. 8). The area explored by a particle with bulk diffusivity $D_0$ is obtained by integration over a

drying cycle $\tau$. The average diffusive distance ($\delta_D$) is then obtained using the yearly number of wetting cycles $N_{cyc}$ (Eqs. 9 and 10).

$$D_e = D_0 \frac{\theta^{\frac{10}{3}}}{\theta_s^2} \tag{8}$$

$$A_{D\tau} = 4\pi \int_0^\tau D_e(\theta, t)dt = 4\pi \frac{D_0 \theta_{FC}^{\frac{10}{3}}}{\theta_s^2} \frac{3}{10\alpha}(1 - e^{\frac{-10}{3}\alpha\tau}) \tag{9}$$

$$\delta_D = \sqrt{N_{cyc} A_{D\tau}} \tag{10}$$

The total number of bacterial cells sustained within the diffusive sphere around POM is estimated considering $N_{POM}$ and $\delta_D$. The population of bacterial cells around a point source (of POM) is assumed to decay radially with exponential rate $\delta_D^{-1}$. Integration over radius $r$ results in the expression for cell density $\rho_c$ that uses number of POM sources $N_{POM}$ and carrying capacity $\rho_{CC}$ (Eqs. 11 and 12).

$$\frac{\rho_c}{\rho_{CC}} = 4\pi N_{POM} \int_0^\infty e^{\frac{-r}{\delta_D}} r^2 dr \tag{11}$$

$$\rho_c = \rho_{CC} 8\pi N_{POM} \delta_D^3 \tag{12}$$

### Conversion of cell densities using soil particle surface area.
Soil bacterial bulk cell density is estimated using soil microbial biomass carbon[2] as previously described[5]. Bacterial cells are mostly attached to particle surfaces in soil[12] and sediments[21,51]. The specific soil-particle surface area ($SSA$) can be estimated using clay content $f_{clay}$ and information on the dominant clay minerals. We consider proportions of kaolinite, illite, and smectite ($f_K, f_I, f_S$) obtained from global maps[42] that dominate most of the soil clay fraction and are each associated with different surface areas[53] ($SA = 60, 200, 590$ m$^2$ g$^{-1}$, respectively). Only a fraction of the soil pore space is considered accessible to bacterial cells. We use 0.4% of the particle surface area[23] ($\eta = 0.0038 \pm 0.0005$, $n = 6$). The following Eq. 13 is used to estimate volumetric $SSA$ ($SSA_v$) using soil bulk density[43] $\rho_{soil}$ with $f_{dom} = f_K + f_S + f_I$.

$$SSA_v = \eta \rho_{soil}(f_{clay}(f_K SA_K + f_S SA_S + f_I SA_I + (1 - f_{dom})\overline{SA}) + 1.1) \tag{13}$$

### Spatially explicit individual-based model (SIM) of bacterial growth on soil grain surfaces.
A spatially explicit individual-based model (SIM) was previously implemented[5,54] and used here with few modifications. Cells of multiple species (characterized by two kinetic parameters) consume carbon sources and move continuously (active swimming and passive shoving) on hydrated heterogeneous soil surfaces. The heterogeneous domain describes pore surfaces within a thin slab of soil (1 by 1 mm) with 10 μm thickness and periodic boundary conditions. The domain is partitioned using a hexagonal grid resulting in grid cells of 100 μm$^2$. The water holding capacity of each grid cell was specified by sampling from a random uniform distribution and assigning a total porosity of 0.5. For different (macroscopic) water contents the total volume of water was partitioned proportional to each grid cells water holding capacity while preserving the prescribed volume of water. For each grid cell the effective water film thickness was calculated (dividing the volume of water by the area of the grid cell) and was used to calculate local diffusive fluxes and the velocity of cellular motion as previously described[5]. In the SIM, a bacterial species is defined by two Monod parameters for each of the three nutrients ($2 \times 3$ parameters per species) prescribed from a range of values[54]. Here, the number of species was reduced compared to previous implementations[5,54] by coarsening the discretization of the cell physiological parameter space that resulted in an initial cell number and richness of 504 cells, one of each species. The model considers a point source of diffusible carbon localized in the center of the warped domain representing 1 mm$^2$ of soil particle surface area with constant concentration boundary condition ($C_A = 25$ g m$^{-3}$). In preliminary simulations, we have tested several concentrations ($0.5 \times C_A$, $C_A$, and $2 \times C_A$) and selected the concentration, which resulted in cell densities comparable to observed values. Three carbon sources ($A$, $B$, and $C$ with assigned yields $Y = 0.25, 0.5$, and $0.75$ g$_{cell}$ g$^{-1}$) represent a metabolic cascade ($A \rightarrow B \rightarrow C$). The carbon source $A$ with the lowest yield was initially provided and carbon mass was conserved by assigning the "leftover" carbon to the next carbon type (e. g. $A \rightarrow B$ with efficiency $1 - Y_A$). The simulations were performed for a range of hydration conditions and a duration of eight days at a one-minute time step. The total cell density, spatial coordinates of the cells, and the species identity were recorded for further analysis.

### Spatial cell aggregation model—cell cluster size distribution.
In the BIHM, an exponentially truncated power law describes cell cluster sizes assuming that individuals tend to aggregate within a finite space[32]. This general distribution can be applied to describe animal group sizes[32], aggregation patterns of bacterial cells[33,34], and the distribution of soil bacterial aqueous habitats[5]. The probability $P(n)$ of having a group of $n$ individuals is described as in Eq. (14) with exponent $b$ and cutoff size $n_c$. The normalization constant $A$ is given by constraint 15 using the largest observed group size $n_{max}$.

$$P(n) = An^{-b} e^{-\frac{n}{n_c}} \tag{14}$$

$$1 = \sum_{1}^{n_{max}} P(n) \tag{15}$$

Similarly, the density fluctuations in growing bacterial colonies[34] and the distribution of bacterial cluster sizes on leaf surfaces[33] follow such cluster statistics where $n_c$ is related to the total number of cells in the system[34]. Also, we can calculate the number of cell clusters $N(n)$ and cells $N_c(n)$ for every size class $n$ using cell density $\rho_C$ and Eqs. (16–18).

$$p(n) = nP(n) = An^{1-b}e^{-\frac{n}{n_c}} \tag{16}$$

$$N(n) = \rho_C \frac{p(n)}{\sum p(n)} \tag{17}$$

$$N_c(n) = nN(n) \tag{18}$$

In soils, we expect $b$ and $n_c$ to vary with aqueous phase connectivity. The relation of both parameters with water contents and cell density is not known a priori and were determined using the simulated cell cluster size distributions obtained from the SIM under varying water contents. The dependency of $n_c$ on cell density $\rho_c$ is described using Eq. (19) (with fitting parameters $a_n, b_n$) in agreement with model data (Fig. S4). Parameters $n_c$ and $b$ are related and Eq. (20) is used for $b$ (with fitting parameters $\alpha, \beta, \gamma$). These relations are used to parametrize the cell cluster size distribution of the BIHM using the SIM results.

$$n_c(\rho_c) = a_n\rho_c^{b_n} \tag{19}$$

$$b(\rho_c) = \begin{cases} 1, & b < 1 \\ \alpha(\frac{\rho_c}{n_c})^{\beta} + \gamma, & b \geq 1 \end{cases} \tag{20}$$

We assume that cell clusters are spread uniformly on the accessible soil surface area to estimate the average distance between bacterial cell clusters. This assumption is justified for soil since the distances between cell clusters is much larger than the size of the clusters. Alternatively, biomass quantiles could be calculated based on the distance to POM by setting appropriate bounds for the integration of Eq. (11) to distinguish, for example, bulk soil cell clusters from those inhabiting rhizosphere "hot spots"[24].

**Cell cluster size-induced anoxic microsites.** Cell clusters can spontaneously deplete oxygen in their core if cell numbers and cell densities are high enough as previously described[36]. The oxygen diffusing trough a dense and compact cell cluster ($10^9$ cells per mm³) can be intercepted by metabolizing cells. We assumed a constant biomass-specific oxygen uptake rate ($k_{O_2} = 46\,\text{g s}^{-1}\,\text{m}^{-3}$), oxygen diffusivity within a cell cluster ($D_{O_2} = 1.12 \times 10^{-9}\,\text{m}^2\,\text{s}^{-1}$) and saturated oxygen concentration in the liquid phase[36] ($S_{O_2} = 8.24\,\text{g m}^{-3}$). For simplicity, spherical cell clusters were considered. Using Eq. (21), we estimated the minimum cell cluster radius that could induce anoxic conditions at the core. Combined with the cluster size distribution obtained from the BIHM we estimated the proportion of biomass associated with such anoxic clusters. For the spatial mapping we use Eq. (12) to estimate cell density and the associated cell cluster sizes. We note that this calculation is based solely on the cluster size distribution of the BIHM and no oxygen limitation was considered in the SIM microscale simulations. The consideration of anoxic microsites induced within sizable cell clusters illustrates cell cluster size effects on soil functioning and demonstrates the utility of collecting microscopic information on bacterial populations.

$$R_{min} = \sqrt{\frac{6D_{O_2}S_{O_2}}{k_{O_2}}} \tag{21}$$

**Soil microcosm experiment.** A nutrient-rich garden soil (on ETH campus, 47°22′43.8″ N and 8°32′53.6″E) was sampled between 5–10 cm depth in March 2018 and subsequently sieved (<2 mm) following air drying for 3 h. The soil was incubated for 4 days at 28 °C on the porous surface model[55] that allowed for controlled hydration conditions. The experimental setup consisted of four hydrated ceramics with three holes drilled in each that were filled with soil (4 mm diameter and 3 mm depth, three replicate samples). Four treatments were applied by independently varying hydration conditions (−35 cm and −5 cm matric potential) and nutrient concentration (autoclaved tap water and tryptic soy broth, TSB). TSB is a general-purpose liquid enrichment media that supports a wide range of bacteria and is also used as a sterility test medium. It is used here because it can support growth of a wide range of (unknown) bacterial species. After incubation, soils were stained following the manufacturer's guidelines using SYTO9 to label DNA (Thermofisher Scientific; 3 µl were applied to each sample with a concentration of 10 µM and incubated for 20 min). For image acquisition, an epifluorescence microscope was used with a GFP filter cube (EVOS FL Auto, Life Technologies, Zug, Switzerland). Imaging was done in situ at the soil surface since the microcosm was small enough to fit on the inverted microscope (turning the ceramic upside down). For each ceramic, at least 9 images (3–4 for each hole) were taken with the stock objective (AMG, 10X LPlan FL PH; AMEP-4681) covering an area of $1167 \times 876\,\mu\text{m}^2$ at a resolution of 0.91 µm. Constant light settings were used throughout the experiment

(light intensity = 10; exposure = 330 ms; gain = 0 dB). Images were taken at a single plane by maximizing the area in focus (autofocus). Staining and imaging were done under suction (−50 cm matric potential) to remove excess water from the soil surface. Measurements were obtained after two and four days.

**Image analysis for determination of cell locations.** The images were analyzed in Python using the SciPy stack[56]. Greyscale images were normalized to the range of pixel intensity (max–min). Images were denoised using (approximately) shift-invariant wavelet denoising (cycle-spinning)[57] as implemented in the skimage function "cycle_spin" with max_shifts = 9 and wavelet denoising[58] implemented in "denoise_wavelet" using the Haar wavelet. Images taken at 10× resolution were used to localize individual cells. First, the area of focus was detected based on singular value decomposition[59] with a window size of 29 and retaining the 3 most significant singular values. The resulting blur map was converted to a binary mask using cross-entropy thresholding ("threshold_li" in skimage[60]) and corresponds to the region in focus (effectively removing parts that contain no information). Holes and objects were removed from the mask if they were smaller than 25 pixels. Positions of cells were detected using the Laplacian of Gaussian method that is capable of sub-resolution edge detection and was implemented in skimage[60]. A range of standard deviations was considered to detect local intensity peaks ($\sigma = 0.4$–7.8 in 40 steps). The smallest object was represented by a standard deviation of 0.5 µm and the largest 10 µm. Coordinates of each cell were only used if they lied within the area of focus as determined by the blur detection. Total cell density was obtained by dividing the total number of cells by the area in focus. The individual steps of image processing are illustrated in Fig. S5. Images at increased resolution were used to confirm labeling of cells in the microcosm experiment (Fig. S6).

**Clustering of proximal cells for estimation of cell cluster size distributions.** Cells within a Euclidean distance of 5 µm were assigned to the same cell cluster. Agglomerative clustering (single linkage) was used where computationally feasible. For cell numbers exceeding 30,000, HDBSCAN[61] was used instead (with similar parameters: min_cluster_size = 3, min_samples = 3, cluster_selection_epsilon = 5 µm, cluster_selection_method = "eom"). The usage of HDBSCAN did not affect the clustering considerably.

Data was pooled to a single cell cluster size distribution to increase the counts of large clusters (that are unlikely to be observed within small areas). Replicate simulations of the SIM were pooled for each water content. For a previous study[9] and our microcosm experiment, images were pooled since no substantial differences in cell density were detected across samples and treatments. The spatial aggregation model (Eq. 14) was fitted to the distribution of cell cluster sizes using maximum likelihood to obtain estimates of $b$ and $n_c$ (as implemented in "powerlaw" [62–64] with parameters discrete = True, discrete_approximation = "xmax" and xmin = 1).

**Reporting summary.** Further information on research design is available in the Nature Portfolio Reporting Summary linked to this article.

## Data availability

All maps generated, the image data and the image analysis code are archived on Zenodo (DOI: 10.5281/zenodo.6325132). Additional data underlying the main graphs is provided as a source data file (Supplementary Data 1).

## Code availability

The code underlying the SIM was previously published (DOI: 10.5281/zenodo.3558542).

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

## Acknowledgements

We gratefully acknowledge Jingyu Wang for assistance with microscopy work and Minsu Kim for discussions on model assumptions and model formulation. Portions of the paper were developed from the doctoral thesis of SB (https://doi.org/10.3929/ethz-b-000453288). This work was funded by the European Research Council (ERC) Advanced Grant SoilLife (No 320499) and MicroScapesX (SystemsX.ch). The work was conducted at ETH Zurich.

## Author contributions

D.O. and S.B. designed the research; D.O. and S.B. performed research; S.B. wrote the computer code; S.B. performed experiments. D.O. and S.B. analyzed results and authored the paper.

## Competing interests

The authors declare that they have no competing interests.
