## [Peer Review File · Communications Biology]

Reviewers' comments:

Reviewer #1 (Remarks to the Author):

This is a very interesting paper, well written, well documented and building on the authors' previous published work on soil bacteria modelling. I commend the considerate effort the authors have put to validate the models they are proposing.

As with every model, especially complex ones as the ones proposed here, critics could use von Neumann argument "With four parameters I can fit an elephant, and with five I can make him wiggle his trunk". Indeed, Figure 3 a) and b) can be in this category, where a model seems to be "made" to match the experiment. Nevertheless, individual-based models are one of the few types of models adequate to represent the distribution of bacterial cells in soils and the authors are proposing a novel way to connect the scales (from micro to sub-millimetre scale through the bacterial interaction heuristic model). Scaling up it is still a challenge in individual based models as it implies simplification and important features explicitly modelled at the level of the individual may be lost when moving towards a community (of bacteria, in this case). By providing experimental data from their microcosm experiments the authors bring evidence that the framework that they propose is suitable for simulating and interpreting data from bulk soil samples.

Reviewer #2 (Remarks to the Author):

The authors propose a framework to predict the spatial distribution of soil bacterial communities at sub-milimeter scales from the biome-specific bacterial cell density, which is determined by carbon inputs across different climate conditions and soil types using an exponentially truncated power law. Interestingly, they find that cell cluster sizes obtained from different independent sources involving experimental and simulation experiments collapse after scaling the data using two a priori unknown parameters that they parametrise using an spatially explicit individual-based model of soil bacteria. As the same authors do in a former published work (reference 5), the manuscript combine elegantly experimentation, available data and modelling strategies including an existing 2D individual-based model and achieve combining multiple temporal and spatial scales likely to inspire further works in the field and further advance on a more mechanistic knowledge of soil functioning.

The methodology requires making many working assumptions, but, to my understanding, the assumptions and expressions used seem sound. The fact that the authors use a 2D instead of a 3D approach in the mechanistic modelling can have an effect on the connectivity of the medium (and thus the cell micro-colonies) connectivity as pointed out recently, for instance, by Schluter et al (2019) Environ. Sci. Technol. 53, 829-837. Nonetheless, the authors of the manuscript being reviewed suggest that the general approach they propose needs to be further tested, and validated, which is compatible with this observation.

I suggest to review the use of "cell density", e.g., using "total cell density", "bulk cell density", or "biome-specific cell density" consistently. It is a key output of the model and provided that the manuscript deals with total population and density of cell clusters, parts of the text are confusing.

More specific comments can be found below:

Ln(s) 22-23. The distribution of soil bacteria and/or bacterial colonies can be imaged despite being time consuming as it is done, for instance, in the work by Xavier Raynaud and Naoise Nunan (2014) already cited. I suggest rephrasing accordingly, for instance, using "difficult to observe" instead of "unobservable", or something on these lines.

Ln(s) 81. It is not clear which parts of the methodology are specifically referred as BIHM model, please clarify.

Ln(s) 121. Is there a reason for using five micrometres to distinguish neighbouring cells? Could you state, please?

Ln(s) 392-393. Please, review these lines of text. Could the text of the subfigure 1 b be grouped together?

Ln(s) 463. Please, add "bacterial" or "microbial" before the word "cells" for an improved clarity.

Ln(s) 463. "maintaining bacterial cells" reads a bit odd. I suggest rephrasing.

Ln(s) 465. How was the exponential rate assumed? Could you specify?

Ln(s) 473. I suggest using directly "(f_k, f_i, f_s)" instead of "(K, I, S)".

Ln(s) 479. SA is not described in the text, only SAK, SAs and SAi are.

Ln(s) 480-493. More details about the SIM model helping to understand how it has been used in the contribution in question should be provided. For instance, the description provided in the reference 5 is much more comprehensive. At least, the formalism used to represent the domain and if the domain is whether homogenous or heterogoneous should be stated. In addition, do the computational domain differ when different

Ln(s) 483. Do all the individuals of the same specie behave the same or variability is taken into account?

Ln(s) 485-86. The number of species used is not explained. Please state.

Ln(s) 487. How is the constant concentration boundary condition chosen? Please explain.

Ln(s) 506. What is $p(n)$? it is not stated in the text.

Ln(s) 510. I understand that the relationship of b and n_c with water contents and carrying capacity is determined using numerical simulations with the SIM model. Making it clearer in the current sentence would make the text more understandable.

Ln(s) 512-13, it is not clear what is the purpose of the sentence, and, in addition, it seems unfinished, please complete.

Ln(s) 536. Please, review, I understand that this equation should be Eq. 21.

Ln(s) 533-535, the links between the case of anoxic microsities and the implication on soil functioning that highlights the relevance of microscopic traits of bacterial populations is not clear. The wording "microscopic traits of bacterial populations" can be confused with the traits of the bacteria making up the subpopulations, but I understand that the sentence refers to the characteristics of the subpopulations.

Ln(s) 590-591. I am not sure of following why the variability in D_0 was not considered. Although understandable given the assumptions taken all over the framework, the stated justification doesn't need to be necessarily a good reason.

Ln(s) 632-637. For an improved clarity, I suggest restating in the text what W and TSB mean.

Point-by-point response to referee's comments for the manuscript entitled "Aqueous habitats and carbon inputs shape the microscale geography and interaction ranges of soil bacteria"

COMMSBIO-22-2901

Samuel Bickel and Dani Or

Point-by-point response to reviewers' comments

Reviewer #1 (Remarks to the Author):

R1.1 *"This is a very interesting paper, well written, well documented and building on the authors' previous published work on soil bacteria modelling. I commend the considerate effort the authors have put to validate the models they are proposing."*

Thank you for the encouraging comment.

R1.2 *"As with every model, especially complex ones as the ones proposed here, critics could use von Neumann argument "With four parameters I can fit an elephant, and with five I can make him wiggle his trunk". Indeed, Figure 3 a) and b) can be in this category, where a model seems to be "made" to match the experiment."*

While we agree, that, excessive parameters are generally a drawback to modeling, in the spatially-explicit individual-based model (SIM) all (bio-) physical parameters were kept constant (e.g., diffusivity, porosity, cell radius, cell mass) and we varied only the unknown growth parameters within a uniform range of values. Thus, any "fitting" can be attributed to boundary conditions (resource concentration) for simulating cell densities found in soil (see also **R2.16**). Consequently, the relations between average cell density (reflecting carbon inputs and water content) and bacterial cell aggregation patterns result strictly from mechanistic processes incorporated in the model (i.e., habitat connectivity affecting growth and motility) and not any other "fitting" effort.

R1.3 *"Nevertheless, individual-based models are one of the few types of models adequate to represent the distribution of bacterial cells in soils and the authors are proposing a novel way to connect the scales (from micro to sub-millimetre scale through the bacterial interaction heuristic model). Scaling up it is still a challenge in individual based models as it implies simplification and important features explicitly modelled at the level of the individual may be lost when moving towards a community (of bacteria, in this case). By providing experimental data from their microcosm experiments the authors bring evidence that the framework that they propose is suitable for simulating and interpreting data from bulk soil samples."*

We also think that individual-based models are suitable for describing the heterogeneous soil habitat and agree that upscaling remains a challenge. Nevertheless, we place great value in mechanistic model results that provide valuable insights into yet unobservable bacterial activity and interactions in soil and can potentially guide future experimentation and exploration of microbial life at small scales.

Reviewer #2 (Remarks to the Author):

R2.1 *“The authors propose a framework to predict the spatial distribution of soil bacterial communities at sub-milimeter scales from the biome-specific bacterial cell density, which is determined by carbon inputs across different climate conditions and soil types using an exponentially truncated power law. Interestingly, they find that cell cluster sizes obtained from different independent sources involving experimental and simulation experiments collapse after scaling the data using two a priori unknown parameters that they parametrise using an spatially explicit individual-based model of soil bacteria. As the same authors do in a former published work (reference 5), the manuscript combine elegantly experimentation, available data and modelling strategies including an existing 2D individual-based model and achieve combining multiple temporal and spatial scales likely to inspire further works in the field and further advance on a more mechanistic knowledge of soil functioning.”*

We thank the reviewer for the concise summary of our work and the many helpful comments that motivated our manuscripts revision. We also think that the study advances the mechanistic knowledge of soil functioning.

R2.2 *“The methodology requires making many working assumptions, but, to my understanding, the assumptions and expressions used seem sound. The fact that the authors use a 2D instead of a 3D approach in the mechanistic modelling can have an effect on the connectivity of the medium (and thus the cell micro-colonies) connectivity as pointed out recently, for instance, by Schluter et al (2019) Environ. Sci. Technol. 53, 829-837. Nonetheless, the authors of the manuscript being reviewed suggest that the general approach they propose needs to be further tested, and validated, which is compatible with this observation.”*

In previous work² we have compared our individual-based model (2D) to a heuristic model based on percolation theory (2D/3D). The changes in connectivity are well described using (dimensionality-specific) scaling exponents of standard percolation theory and we could reconcile differences in habitat structure between the two models. Briefly, for 3D pore networks, lower water contents are required for the aqueous phase to be connected (i.e., to obtain a spanning aqueous cluster) compared to 2D. How the distribution of the liquid phase affects the distribution of cells requires further research, especially regarding the biological variability (e.g., bacterial morphological traits) and soil boundary conditions (e.g., directional air/liquid invasion). Nonetheless, our approach strikes a balance between simplicity and generality by capturing salient features of the strongly nonlinear effects of aqueous phase connectivity on (local) resource fluxes and overall bacterial cell density in soil.

R2.3 *“I suggest to review the use of “cell density”, e.g., using “total cell density”, “bulk cell density”, or “biome-specific cell density” consistently. It is a key output of the model and provided that the manuscript deals with total population and density of cell clusters, parts of the text are confusing.”*

Thank you for pointing out the ambiguous use of cell density. We have changed the phrasing as suggested and now use “bulk cell density” when referring to (soil) sample-scale and biome-specific cell density. We use “cell density” in general statements (independent of spatial scale) and “average cell density” when referring to averages of small-scale cell density (i.e., as obtained from simulations and experimental observations).

R2.4 “More specific comments can be found below: Ln(s) 22-23. The distribution of soil bacteria and/or bacterial colonies can be imaged despite being time consuming as it is done, for instance, in the work by Xavier Raynaud and Naoise Nunan (2014) already cited. I suggest rephrasing accordingly, for instance, using “difficult to observe” instead of “unobservable”, or something on these lines.”

Thank you for the specific comments that improved the clarity of the manuscript. We agree that the phrasing was inaccurate and changed the sentence to “Micro-geographic considerations of difficult-to-observe microbial processes can improve the interpretation of data from bulk soil samples.” (L 21-23).

R2.5 “Ln(s) 81. It is not clear which parts of the methodology are specifically referred as BIHM model, please clarify.”

We have clarified in the main text (L 72-74) that the *bacterial interaction heuristic model* (BIHM) refers to the analytical formulation that describes biome-specific bacterial bulk cell density and the links to the microscale cell cluster size distribution (Equations 12, 14, 19, and 20).

R2.6 “Ln(s) 121. Is there a reason for using five micrometres to distinguish neighbouring cells? Could you state, please?”

We thank the reviewer for raising this important point. Simply stated, we have chosen a value that permits cells to interact mechanically (or physically via appendages) while interactions over larger distances would require either active motility or diffusion mediated processes. The distance of five micrometers was guided by results from a previous study³ that measured a value of 5.4 μm based on pair-correlations where cell cluster size distribution was robust against modest changes in distance to neighboring cells. Additionally, we considered this distance to be reachable by cell division within a few generations (1-3 divisions) without active motility. This separation distance has been shown to be relevant for cell-cell interactions via nanotubes⁴ that can extend to several μm (covering 9 μm^2 in *B. subtilis* under low cell densities⁴). We agree that a different value can be selected based on the nature of the research questions.

R2.7 “Ln(s) 392-393. Please, review these lines of text. Could the text of the subfigure 1 b be grouped together?”

We have corrected two mistakes in the figure caption concerning the panel labeling. Panel **d** was mislabeled as panel **e**, and panel **e** was mislabeled as panel **b**.

R2.8 “Ln(s) 463. Please, add “bacterial” or “microbial” before the word “cells” for an improved clarity.”

Amended.

R2.9 “Ln(s) 463. “maintaining bacterial cells” reads a bit odd. I suggest rephrasing.”

The phrasing was adapted to: “The population of bacterial cells around a point source (of POM) is assumed [...]”.

R2.10 “Ln(s) 465. How was the exponential rate assumed? Could you specify?”

The rate is assumed based on the diffusive distance (δ_D) a small molecule could travel under climatic hydration conditions considering yearly turnover of POM. We assume that cells intercept most nutrients for cellular maintenance within δ_D .

R2.11 “Ln(s) 473. I suggest using directly “(fk, fi, fs)” instead of “(K, I, S)”.”

Amended.

R2.12 “Ln(s) 479. SA is not described in the text, only SAK, SAs and SAi are. “

Amended.

R2.13 “Ln(s) 480-493. More details about the SIM model helping to understand how it has been used in the contribution in question should be provided. For instance, the description provided in the reference 5 is much more comprehensive. At least, the formalism used to represent the domain and if the domain is whether homogenous or heterogoneous should be stated. In addition, do the computational domain differ when different “

We added additional details to the model description. The formalism used to present the heterogeneous domain was added to the methods section (L 480-489).

R2.14 “Ln(s) 483. Do all the individuals of the same specie behave the same or variability is taken into account?”

Each individual’s behavior depends on the local environment (water film thickness and nutrient concentrations in every grid cell). However, the resource usage is deterministic (i.e., an individual of the same species has the same uptake rates for the same local nutrient concentrations). The active motility (swimming) depends on the gradients in the nutrient fields with a random component and is thus not fully deterministic (i.e., it is different for every individual).

R2.15 “Ln(s) 485-86. The number of species used is not explained. Please state.”

In this work, we define each bacterial species via two Monod parameters (maximum specific growth rate and substrate affinity) for each nutrient source. Hence, a bacterial species is defined by a set of six parameters in the simulations for this study (μ_{max} and K_s for three carbon sources). We assigned ranges of parameters to obtain all combinations that are possible. Depending on the discretization of the parameter space we obtain the number of species. Briefly, we obtain 504 combinations of 2x3 parameters, which uniquely define a species. From each species, one cell is randomly placed onto the modelled domain. We added the explanation at L 488-491.

R2.16 “Ln(s) 487. How is the constant concentration boundary condition chosen? Please explain.”

As stated in **R1.2**, the concentration boundary condition was a tuning parameter based on the obtained cell densities. We selected a concentration that resulted in a range of cell densities comparable to empirical estimates in soils of different biomes based on observations and on simple calculations of soil carrying capacity assuming minimal growth and primarily maintenance rates (see Supplementary Text). An explanation has been added to the model description (L 495-496).

R2.17 “Ln(s) 506. What is $p(n)$? it is not stated in the text.”

We have added the definition of $p(n)$ to the equation. It comes from multiplying n with $P(n)$.

R2.18 “Ln(s) 510. I understand that the relationship of b and nc with water contents and carrying capacity is determined using numerical simulations with the SIM model. Making it clearer in the current sentence would make the text more understandable.”

We have clarified in the text that the parameters were determined using the simulated cell cluster size distributions obtained from the SIM under varying water contents (L 517-519).

R2.19 “Ln(s) 512-13, it is not clear what is the purpose of the sentence, and, in addition, it seems unfinished, please complete.”

The incomplete sentence was not needed and was removed.

R2.20 “Ln(s) 536. Please, review, I understand that this equation should be Eq. 21.”

Thank you for pointing out this mistake. The equation number has been corrected.

R2.21 “Ln(s) 533-535, the links between the case of anoxic microsites and the implication on soil functioning that highlights the relevance of microscopic traits of bacterial populations is not clear. The wording “microscopic traits of bacterial populations” can be confused with the traits of the bacteria making up the subpopulations, but I understand that the sentence refers to the characteristics of the subpopulations.”

The sentence was unclear. It refers to properties of the cell cluster (population) sizes and we have changed the phrasing to: “The consideration of anoxic microsites induced within sizable cell clusters illustrates cell cluster size effects on soil functioning and demonstrates the utility of collecting microscopic information on bacterial populations.”

R2.22 “Ln(s) 590-591. I am not sure of following why the variability in D_0 was not considered. Although understandable given the assumptions taken all over the framework, the stated justification doesn't need to be necessarily a good reason.”

The reasoning is based on the analysis of the governing equations. The diffusivity D_0 of small molecules through liquid with low Reynolds numbers varies as $D_0 \sim \frac{T}{r}$ with temperature T and radius of the (spherical) molecule r (Einstein-Stokes equation). For both properties (size of the molecule and temperature) the exponent is 1. On the other hand, the effective solute diffusivity D_e for different soil water contents varies as $D_e \sim \theta^{\frac{10}{3}} / \theta_s^2$ (Millington-Quirk equation). Hence, for a sufficiently narrow range of T and r , the variation in D_e is driven primarily by changes in soil water content θ (and porosity θ_s).

R2.23 “Ln(s) 632-637. For an improved clarity, I suggest restating in the text what W and TSB mean.”

The information on the experimental conditions was added to the figure legend.

References

1. Tecon, R. & Or, D. Bacterial flagellar motility on hydrated rough surfaces controlled by aqueous film thickness and connectedness. *Sci. Rep.* **6**, 19409 (2016).
2. Bickel, S. & Or, D. Soil bacterial diversity mediated by microscale aqueous-phase processes across biomes. *Nat. Commun.* **11**, 1-9 (2020).
3. Zhang, H. P., Be'er, A., Florin, E.-L. & Swinney, H. L. Collective motion and density fluctuations in bacterial colonies. *Proc. Natl. Acad. Sci.* **107**, 13626-13630 (2010).
4. Dubey, G. P. *et al.* Architecture and Characteristics of Bacterial Nanotubes. *Dev. Cell* **36**, 453-461 (2016).

REVIEWERS' COMMENTS:

Reviewer #1 (Remarks to the Author):

The authors have thoroughly answer to reviewers' comments and this is an improved submission.

Reviewer #2 (Remarks to the Author):

I would like to thank the authors for addressing my concerns. Clarity of the manuscript has been improved in the revised version.

I think my concerns have been addressed and I would recommend accepting the manuscript. I only have 2 minor suggestions that should be easily addressed and that, in my opinion, would further improve clarity of the manuscript.

(I am only copy pasting the relevant text:)

1. R2.5 "Ln(s) 81. It is not clear which parts of the methodology are specifically referred as BIHM model, please clarify."

We have clarified in the main text (L 72-74) that the bacterial interaction heuristic model (BIHM) refers to the analytical formulation that describes biome-specific bacterial bulk cell density and the links to the microscale cell cluster size distribution (Equations 12, 14, 19, and 20).

Reviewer comment: For an improved clarity, I suggest explicitly stating the equations mentioned above in the main text (L 74 of the new manuscript). They are mentioned in the reply but not included in the lines 72-74 of the new manuscript.

2. R2.15 "Ln(s) 485-86. The number of species used is not explained. "

Author reply: In this work, we define each bacterial species via two Monod parameters ... Briefly, we obtain 504 combinations of 2x3 parameters, which uniquely define a species. From each species, one cell is randomly placed onto the modelled domain. We added the explanation at L 488-491

Reviewer reply: From your reply I understand you used one individual of each one of the 504 combinations of parameters (i.e., species) but this is not clear in the manuscript which states (Line 492) "... the cell physiological parameter space that resulted in an initial cell number and richness of 504 cells of each species". I suggest rephrasing so it is clearer, for instance, changing "504 cells of each species" to "504 cells, one of each species".

Point-by-point response to referee's comments for the manuscript entitled "Aqueous habitats and carbon inputs shape the microscale geography and interaction ranges of soil bacteria"

Final revisions for manuscript COMMSBIO-22-2901A

Samuel Bickel and Dani Or

Point-by-point response to reviewers' comments

Reviewer #1 (Remarks to the Author):

R 1.1 *"The authors have thoroughly answer to reviewers' comments and this is an improved submission."*

We thank the reviewer for the feedback that helped improve the manuscript.

Reviewer #2 (Remarks to the Author):

R2.1 *"I would like to thank the authors for addressing my concerns. Clarity of the manuscript has been improved in the revised version. I think my concerns have been addressed and I would recommend accepting the manuscript. I only have 2 minor suggestions that should be easily addressed and that, in my opinion, would further improve clarity of the manuscript. (I am only copy pasting the relevant text:)"*

We thank the reviewer for the insightful comments that improved the clarity of our manuscript. We revised the text as suggested below.

R2.2 *'1. R2.5 "Ln(s) 81. It is not clear which parts of the methodology are specifically referred as BIHM model, please clarify." We have clarified in the main text (L 72-74) that the bacterial interaction heuristic model (BIHM) refers to the analytical formulation that describes biome-specific bacterial bulk cell density and the links to the microscale cell cluster size distribution (Equations 12, 14, 19, and 20). Reviewer comment: For an improved clarity, I suggest explicitly stating the equations mentioned above in the main text (L 74 of the new manuscript). They are mentioned in the reply but not included in the lines 72-74 of the new manuscript.'*

The equation numbers have now been added to the text as suggested.

R2.3 *'2. R2.15 "Ln(s) 485-86. The number of species used is not explained." Author reply: In this work, we define each bacterial species via two Monod parameters ... Briefly, we obtain 504 combinations of 2x3 parameters, which uniquely define a species. From each species, one cell is randomly placed onto the modelled domain. We added the explanation at L 488-491 Reviewer reply: From your reply I understand you used one individual of each one of the 504 combinations of parameters (i.e., species) but this is not clear in the manuscript which states (Line 492) "... the cell physiological parameter space that resulted in an initial cell number and richness of 504 cells of each species". I suggest rephrasing so it is clearer, for instance, changing "504 cells of each species" to "504 cells, one of each species".*

Yes, this is correct. One individual of each species was initialized. We have clarified this as suggested.